# Overexpression of PD-1 and CD39 in tumor-infiltrating lymphocytes compared with peripheral blood lymphocytes in triple-negative breast cancer

Asmaa M. Zahran[1‡], Amal Rayan[2‡*], Zeinab Albadry M. Zahran[3], Wael M. Y. Mohamed[4,5], Dalia O. Mohamed[6], Mona H. Abdel-Rahim[7], Omnia El-Badawy[7‡]

1 Department of Clinical Pathology, South Egypt Cancer Institute, Assiut University, Assiut, Egypt, 2 Clinical Oncology Department, Faculty of Medicine, Assiut University, Assiut, Egypt, 3 Department of Clinical Pathology, Faculty of Medicine, Assiut University, Assiut, Egypt, 4 Oncology Department, Faculty of Medicine, Port Said University, Port Said, Egypt, 5 Consultant Medical Oncologist Locum, Swansea University Hospital, Swansea, United Kingdom, 6 Department of Radiation Oncology, South Egypt Cancer Institute, Assiut University, Assiut, Egypt, 7 Department of Medical Microbiology and Immunology, Faculty of Medicine, Assiut University, Assiut, Egypt

‡ These authors are joint senior authors on this work.
* amalrayan@aun.edu.eg

**Data Availability Statement:** All data are included within the paper.

## Abstract

### Background and aim

Growing evidence highlighted the primary role of the immune system in the disease course of triple-negative breast cancer (TNBC). The study aim was to investigate the expression of PD-1 and CD39 on CD4[+] and CD8[+] cells infiltrating tumor tissue compared to their counterparts in peripheral blood and explore its association with tumor characteristics, disease progression, and prognosis in females with TNBC.

### Patients and methods

The study included 30 TNBC patients and 20 healthy controls. Cancer and normal breast tissue and peripheral blood samples were collected for evaluation of the expression of PD-1 and CD39 on CD4[+] and CD8[+]T cells by flow cytometry.

### Results

A marked reduction in the percentage of CD8[+] T lymphocytes and a significant increase in the frequencies of CD4[+] T lymphocytes and CD4[+] and CD8[+] T lymphocytes expressing PD1 and CD39 were evident in tumor tissue in comparison with the normal breast tissue. The DFS was inversely related to the cancer tissue PD1[+]CD8[+] and CD39[+]CD8[+] T lymphocytes. Almost all studied cells were significantly increased in the tumor tissue than in peripheral blood. Positive correlations were detected among peripheral PD1[+]CD4[+]T lymphocytes and each of cancer tissue PD1[+]CD4[+], PD1[+]CD8[+]and CD39[+]CD8[+]T cells, and among peripheral and cancer tissue CD39[+]CD4[+]and CD39[+]CD8[+] T cells.

**Funding:** the authors received no specific funding for this work.

**Competing interests:** the authors have declared that no competing interests exist

## Conclusions

The CD39 and PD1 inhibitory pathways are synergistically utilized by TNBC cells to evade host immune response causing poor survival. Hence, combinational immunotherapy blocking these pathways might be a promising treatment strategy in this type of cancer.

## Introduction

Globally, an estimated 2 million breast cancer cases were diagnosed by the end of 2018 [1]. Of these, 12%-20% are of a triple-negative phenotype [2]. Triple-negative breast cancer (TNBC) refers to breast cancer phenotype where the estrogen and progesterone receptors are negative [by immunohistochemistry (IHC)] with lack of overexpression of human epidermal growth factor receptor 2 (HER2) [by IHC], or the absence of HER2 neu amplification [by fluorescence *in situ* hybridization technique] [3].

TNBC is a biologically aggressive disease with ineffective treatments; several efforts have been recently conducted to increase therapeutic opportunities. Over the past decade, growing evidence has highlighted the primary role of the immune system in influencing the disease course of TNBC. The presence of a high number of cytotoxic (CD8[+]) tumor-infiltrating lymphocytes (TILs) can define TNBC patients with a better prognosis following neoadjuvant chemotherapy [4, 5]. Furthermore, CD8[+] TILs were correlated with overall survival and progression-free survival in TNBC patients treated with atezolizumab and nab-paclitaxel in Impassion 130 trial [6].

Along with TILs, is the expression of immune evasion molecules in the tumor microenvironment, including programmed death ligand-1 (PD-L1) and PD-L2 in tumor tissue and PD-1 (a cell membrane protein of the CD28 superfamily) on CD4[+] and CD8[+] TILs that have been demonstrated to influence the prognosis of TNBC [7–10]. After binding, PD-1 suppresses tumor cytotoxicity through down-regulation of TILs responses and putting out tumor immunity [11]. A previous study reported high TIL density, PD-L1 expression, and mutation rate in TNBC, compared with other subtypes. Although, the association between PD-L1 expression and prognosis of TNBC remains controversial, the presence of high PD-L1+ TILs ($> 50\%$) was also independently associated with better survival: $HR_{high-PD-L1+TILs} = 0.27$; 95% CI = 0.10–0.69. Accordingly, the presence of PD-L1+ TNBC cells were associated with improved survival, but the results were not statistically significant [12].

Pembrolizumab is a monoclonal antibody with a high affinity and selectivity for PD-1 and approved for the treatment of metastatic TNBC after Phase 3 (KEYNOTE-355) data showed that pembrolizomab in combination with chemotherapy reduced the risk of death by 35% in previously untreated metastatic TNBC patients whose tumors expressed high levels of PD-L1. Atezolizumab is also approved for metastatic TNBC with PD-L1 expression [13].

Extracellular adenosine triphosphate (ATP) released by dying tumor cells is known to boost immune responses in the tumor microenvironment but might also directly kill adjacent tumor cells. CD39 is the rate-limiting ecto-nucleotidase on immune cells. It catalyzes the sequential hydrolysis of pro-inflammatory adenosine triphosphate (ATP) and adenosine diphosphate (ADP), promoting the synthesis of immune-inhibitory adenosine (CD39 hydrolyses ATP into AMP, while CD73 hydrolyses AMP into adenosine), thus limiting the inflammation and suppressing the immune system. This immunosuppressive pathway enhances tumor growth and guards cancer cells [14, 15]. CD39 acts as a strategic molecule in tumor immunity [16] CD39 is expressed on the surface of endothelial cells, neutrophils, monocytes, macrophages, dendritic cells, B cells, and on some T cell and natural killer (NK) cell subsets [17]. Moreover, some

recent research has revealed that CD39 is also expressed by Tregs [18], Th17 cells [19], γδ T cells [20], and Bregs [21] that often infiltrate solid tumors [22].

Through this study, we investigated the expression of PD-1 and CD39 on CD4+ and CD8+cells infiltrating tumor tissue compared to their counterparts in peripheral blood. We also explored their association with tumor characteristics, disease progression, and prognosis in females with TNBC.

## Patients and methods

Thirty females with early TNBC were recruited in this case-controlled study that was carried out in South Egypt Cancer Institute and Assiut university hospital, Faculty of Medicine, Assiut University. Additionally, the study included 20 healthy blood donors as a control group.

### Clinical methodology

A thorough history, physical examination, sonomammography, chest computed tomography (CT), abdominal CT, bone scan, and tumor marker tests (CA15-3, CEA) were made to exclude distant metastasis and to stage the patients. Fresh tumor and adjacent normal breast tissue specimens were collected immediately after surgical treatment with either modified radical mastectomy or breast conservative surgery. IHC findings of estrogen, progesterone, and HER2 receptors were used to select the TNBC cases. The clinico-pathological parameters evaluated for each patient included age, tumor size, lymph node stage, pathologic type, histological grade, type of surgery, and local recurrence. Patients with neoadjuvant chemotherapy or previous history of cancer treatment were excluded. Moreover, patients were treated with chemotherapy with anthracyclines/taxanes-based regimens that were commonly used, postoperative radiotherapy by 3D-technique was delivered whenever indicated, and patients with BCS received 3DRT. In contrast, those with MRM received RT if indicated. Patients' follow-up was used to assess their survival status and local recurrence, which involved recurrences developing in the same quadrant and of similar molecular phenotype. The period between confirmed diagnosis and death due to breast cancer or the last follow-up in surviving patients was applied for disease-free survival (DFS) analysis. Peripheral blood samples were collected from breast cancer patients after surgery, in addition to those obtained from healthy blood donors as controls. All tissue specimens and blood samples were collected before any chemotherapy, radiotherapy, or other treatment that may affect the immune status.

### Sample collection and processing

After providing informed consent, fresh cancer tissue specimens were collected immediately after surgical resection of primary breast cancer. Besides, 20 healthy tissue samples were also taken from areas adjacent to the safety margins of the same patients. The cell suspension was prepared from the tumor tissue by sequential mechanical disruption. To ensure single-cell suspension and remove clumps or debris, the suspension was filtered through cell strainers [23]. The resulting suspension was then incubated with lysing solution for 5 minutes at 4˚C to lyse the contaminating RBCs and was then washed twice with PBS. Besides, two ml peripheral blood samples were collected from both patients and healthy controls for flow cytometry.

### Flow cytometric detection of PD-1 and CD39 expression by T lymphocytes in peripheral blood and breast tissue samples

A one mm³ of the breast tissue in 100 μl of PBS or 50 μl of the blood sample were stained with 5 μl of each of CD8, CD4, CD3 PD-1 and CD39 monoclonal antibodies. All monoclonal

antibodies were purchased from Becton Dickinson (BD) Biosciences, San Jose, CA, USA. After incubation for 20 minutes at 4˚C in the dark, red blood cells (RBCs) lysis was done, then washing with phosphate-buffered saline (PBS). The cells were suspended in PBS, and a total of 50,000 events were acquired for each sample and analyzed by FACSCalibur flow cytometry by Cell Quest software (BD Biosciences, USA). Human IgG was used as an isotype-matched negative control for each sample. A Forward and side scatter histogram was used to define the lymphocytes population. Then, the percentages of PD-1$^+$and CD39$^+$cells were assessed within each of the CD4$^+$ and CD8$^+$ T lymphocytes (Fig 1).

## Statistical analysis

The Statistical Package for Social Sciences, version 26.0 (IBM SPSS, USA) was accomplished for the statistical analysis. Data were represented as mean ± standard deviation (SD) or standard error (SE). by Shapiro-Wilk test some data were normally distributed including peripheral and tissue CD4+ and CD8+, T-lymphocytes, and age. While PD1+ in peripheral and tissue CD4+ and CD8+ (p-values 0.001, 0.005, 0.003, & <0.0001 respectively), also CD39 in peripheral and tissue CD4+, and CD8+ (p-values 0.002, 0.031, 0.001, & 0.02 respectively), and DFS (p = 0.008) were not normally distributed. Independent sample t-test and paired sample t-test were applied to evaluate the statistical differences between normally distributed groups, while Mann-Whitney U-test and Wilcoxon signed ranks tests for non-normally distributed data. Associations between scale variables were explored using Pearson's and Spearman rho correlations. The disease-free survival (DFS) was estimated by Cox regression test. A *p*-value of less than 0.05 was considered significant.

## Results

### Patients' clinico-pathological features

The median age of the patients was 50±11.5 years (range; 28-77 years). The tumor size reached T2 in 63.3% and T3 in 23.3% of the patients. Each of the N0 or N1 lymph node stages was detected in 46.7% of the patients. Most of the patients had infiltrating ductal carcinoma (IDC) (98.3%) and were of G2 (83.3%). Modified radical mastectomy (MRM) was the primary surgical treatment in 73.3%, followed by breast conservative surgery (BCS) in 23.3% of the patients. Meanwhile, 16.7% of them had a local recurrence (Table 1). The mean DFS ±SE was 24.12±1.9 (95% CI = 20.3–27.9 months), and the median DFS ±SE was 19.8±2 (95% CI = 15.9–23.7months), Fig 2.

### Differential expression of PD1 and CD39 on T lymphocytes in tumor and normal breast tissue of patients with TNBC

A marked reduction in the percentage of CD8$^+$ T lymphocytes was detected in tumor tissue compared with healthy breast tissue (*p* = 0.007). Conversely, a significant increase in the frequencies of T lymphocytes, CD4$^+$T lymphocytes, and CD4$^+$ and CD8$^+$ T lymphocytes expressing PD1 and CD39 was evident in tumor tissue in comparison with the healthy breast tissue (Table 2).

### Relations between tumor-infiltrating lymphocytes and the clinico-pathologic characteristics of patients with TNBC

No significant correlations were observed between the mean percentages of different immune cells and most of the different clinico-pathologic parameters of TNBC. The DFS was inversely related to the levels of tumor-infiltrating PD1$^+$CD8$^+$ and CD39$^+$CD8$^+$ T lymphocytes

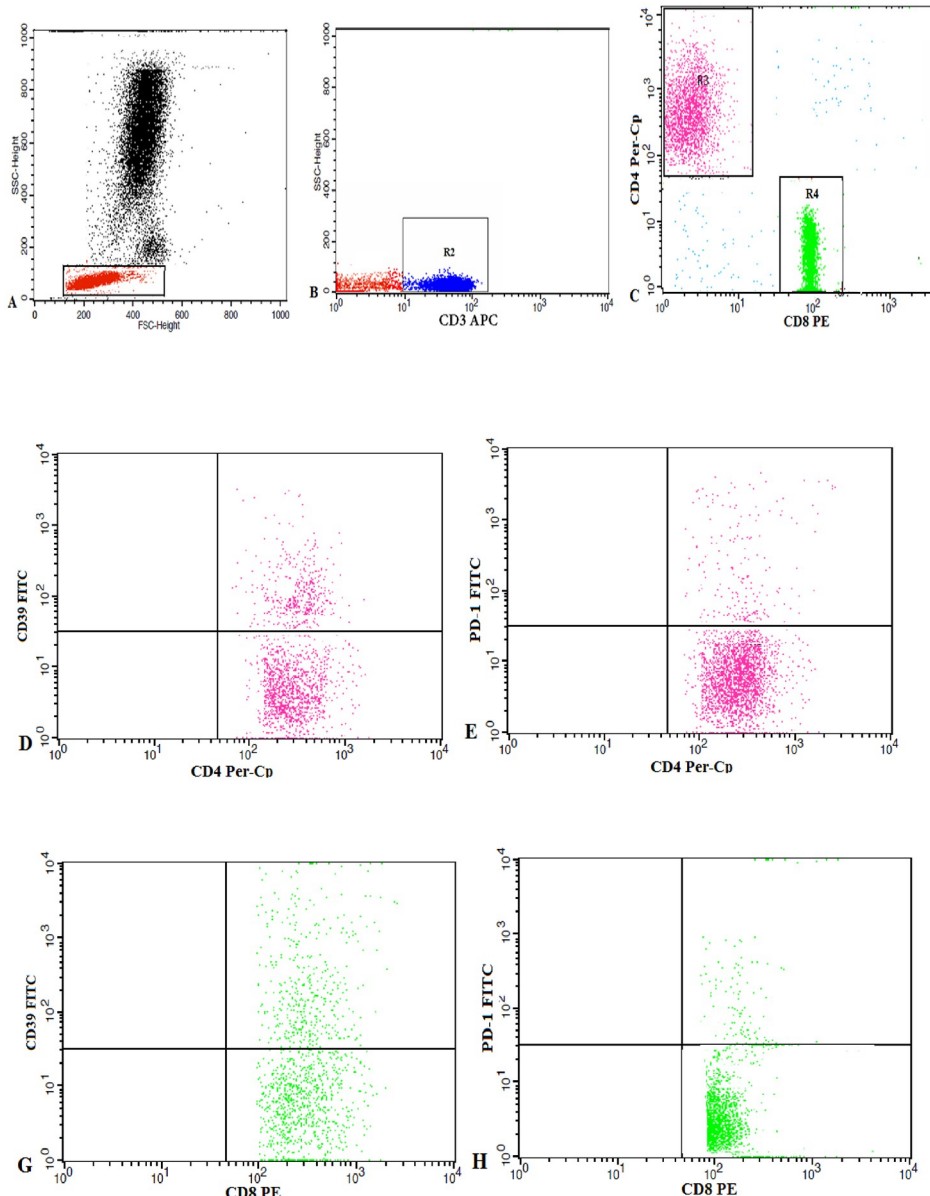

**Fig 1. Flow cytometric analysis of CD39 and PD1 expression by the T Lymphocyte subsets.** A: The lymphocyte population was gated from peripheral blood mononuclear cells by drawing (R1) based on the forward/side scatter characteristics. B, C: CD3 expression was analysed on the gated lymphocytes, and the CD3$^+$ cells were selected by (R2) for further analysis based on CD4 and CD8 expression. (R3) and (R4) represent the regions used to choose the CD4$^+$ andCD8$^+$T cells, respectively. D-H: Dot plots representing CD39 and PD1 expression by CD4$^+$ andCD8$^+$ T cells.

(r = -0.4, $p$ = 0.007 and r = -0.4, $p$ = 0.03 respectively) (Fig 3). A comparison between the levels of the TILs in relation to the different clinicopathologic parameters revealed a significant accumulation of CD39$^+$CD4$^+$T lymphocytes in patients with G2 than G3 ($p$ = 0.002). Tumor-infiltrating PD1$^+$CD4$^+$ T lymphocytes were higher in patients with local recurrence ($p$ = 0.03) (Fig 4).

**Table 1. Clinico-pathological parameters of 30 females with TNBC.**

| Characteristics | Frequency |
|---|---|
| **Age** | |
| median± SD | 50±11.5 |
| Range | 28–77 years |
| **Tumor size** | |
| T1 | 4 (13.3%) |
| T2 | 19 (63.3%) |
| T3 | 7 (23.3%) |
| **Lymph node stage** | |
| N0 | 14 (46.7%) |
| N1 | 14 (46.7%) |
| N3 | 2 (6.7%) |
| **Pathologic type** | |
| IDC | 28 (98.3%) |
| ILC | 2 (6.7%) |
| **Histological grade** | |
| G1 | 1 (3.3%) |
| G2 | 25 (83.3%) |
| G3 | 4 (13.3%) |
| **Surgery** | |
| MRM | 22 (73.3%) |
| BC | 8 (23.3%) |
| **local recurrence** | 5 (16.7%) |

TNBC; triple-negative breast cancer, IDL; infiltrating ductal carcinoma, ILC; infiltrating lobular carcinoma, G; grade, MRM; modified radical mastectomy, BCS; breast conservative surgery, data expressed as mean ± SD or number (percentage).

## Differential expression of PD1 and CD39 on peripheral blood T lymphocytes among patients with TNBC and controls

The percentage of CD8$^+$ T lymphocytes was significantly higher, and that of T lymphocytes and CD4$^+$T lymphocytes was markedly lower in patients in comparison with the healthy controls. The expression of PD1 and CD39 on peripheral blood T cell subsets was elevated in patients than controls, but this difference was not significant among CD39$^+$CD8$^+$ T lymphocytes ($p$ = 0.4) (Table 3).

## Relations of peripheral blood T lymphocytes with the clinicopathologic characteristics of patients with TNBC

Our results illustrated significant negative correlation between PD1$^+$CD8$^+$T lymphocytes and DFS (r = -0.3, $p$ = 0.04) (Fig 5). Comparisons between the levels of the peripheral T lymphocyte subsets in relation to the different clinicopathologic parameters revealed that the mean percentage of PD1$^+$CD8$^+$T lymphocytes in patients with ILC was exceeding that of patients with IDC ($p$ = 0.03). On the contrary, CD39$^+$CD8$^+$T lymphocytes were higher in patients with IDC ($p$ = 0.04), (Fig 6). A lower level of CD39$^+$CD4$^+$ T lymphocytes was detected in G3 TNBC compared with G2 ($p$ = 0.007).

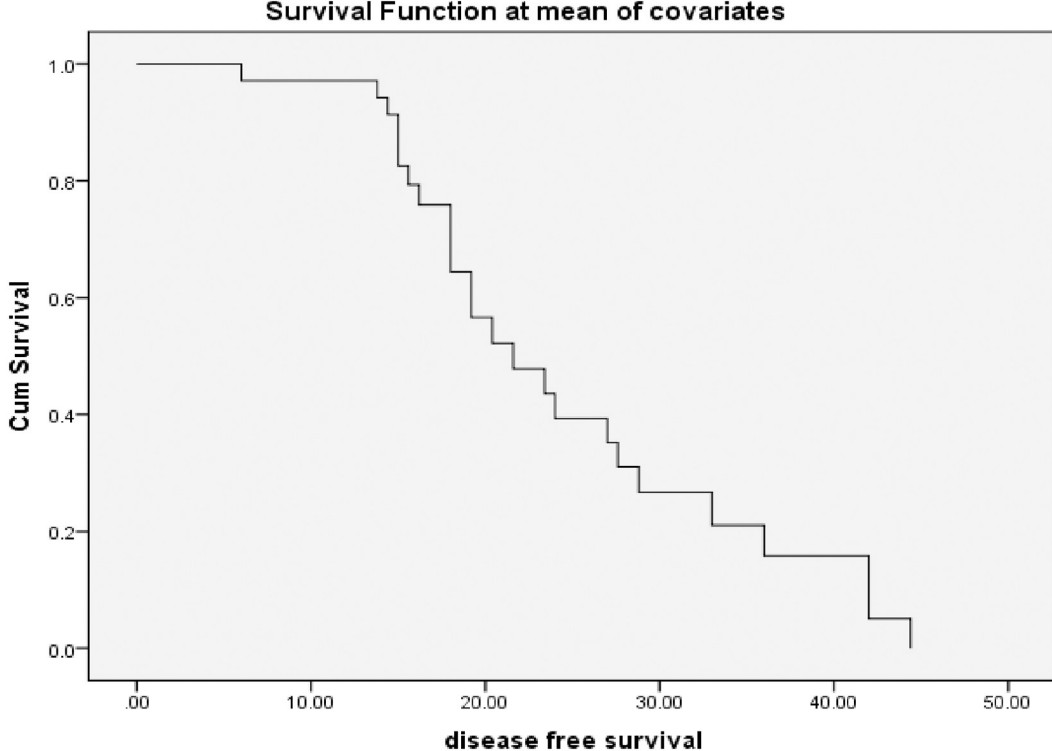

**Fig 2. Disease-free survival (DFS) among 30 patients with triple-negative breast cancer (TNBC).**

## Comparison between the levels of expression of PD1 and CD39 among tumor-infiltrating and peripheral blood lymphocytes of patients with TNBC

All studied cells were significantly increased in the tumor tissue than in peripheral blood, as shown in Table 4, except T lymphocytes, which were significantly lower in tumor tissue than in the peripheral blood.

**Table 2. Differential expression of PD1 and CD39 on T lymphocytes in tumor and normal breast tissue of patients with triple-negative breast cancer.**

| T lymphocyte subsets (%) | Tumor tissue (n = 30) | Normal tissue (n = 20) | p- value |
|---|---|---|---|
| T lymphocytes | 26.6±0.9 | 6.2±0.3 | <**0.0001** |
| CD4[+] | 58.2±3 | 46±2 | **0.002** |
| CD8[+] | 34.2±3 | 44.8±3 | **0.007** |
| PD1[+]CD4[+] | 50.8±4 | 16.9±2 | <**0.0001**[a] |
| PD1[+]CD8[+] | 34.9±2 | 20.6±2 | <**0.0001**[a] |
| CD39[+]CD4[+] | 21±2 | 6.1±0.8 | <**0.0001**[a] |
| CD39[+]CD8[+] | 13.6±0.9 | 3.6±0.3 | <**0.0001**[a] |

The percentages of the CD4[+] and CD8[+] cells were calculated within the T lymphocytes, and the percentages of PD-1[+]and CD39[+]cells were assessed within each of the CD4[+] and CD8[+] T lymphocytes. Data expressed as mean ±SE, Independent samples t-test,

[a]; Mann-Whitney test, a p-value is significant if < 0.05.

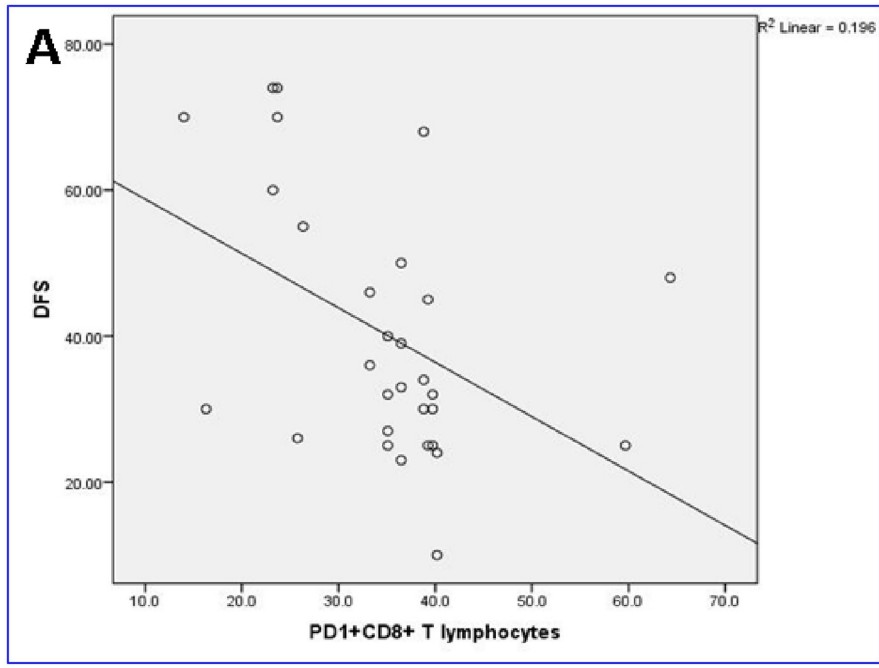

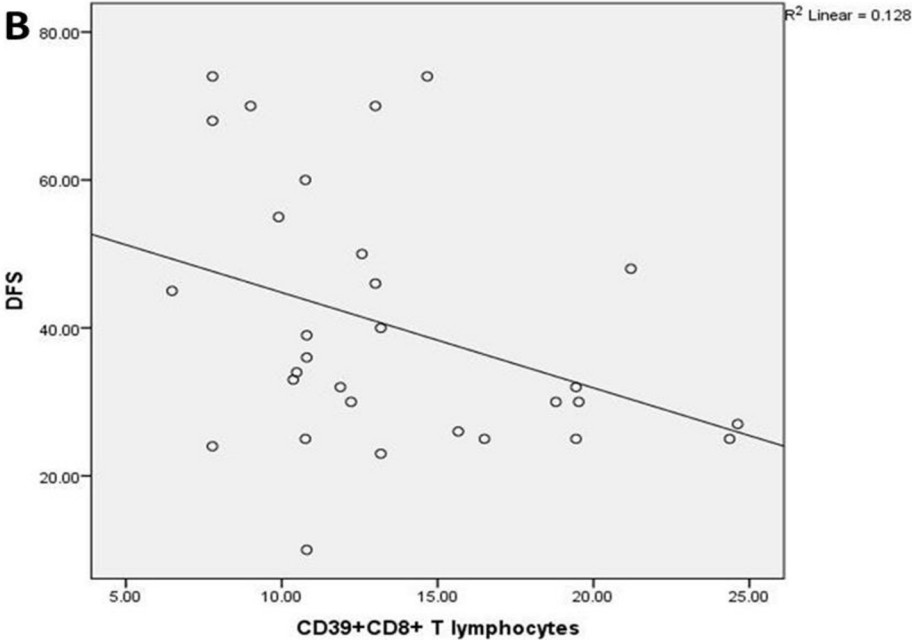

**Fig 3. DFS association with tumor-infiltrating (A) PD1$^+$CD8$^+$ T lymphocytes ($p$ = 0.007), and (B) CD39$^+$CD8$^+$ T lymphocytes ($p$ = 0.03), in 30 TNBC patients.**

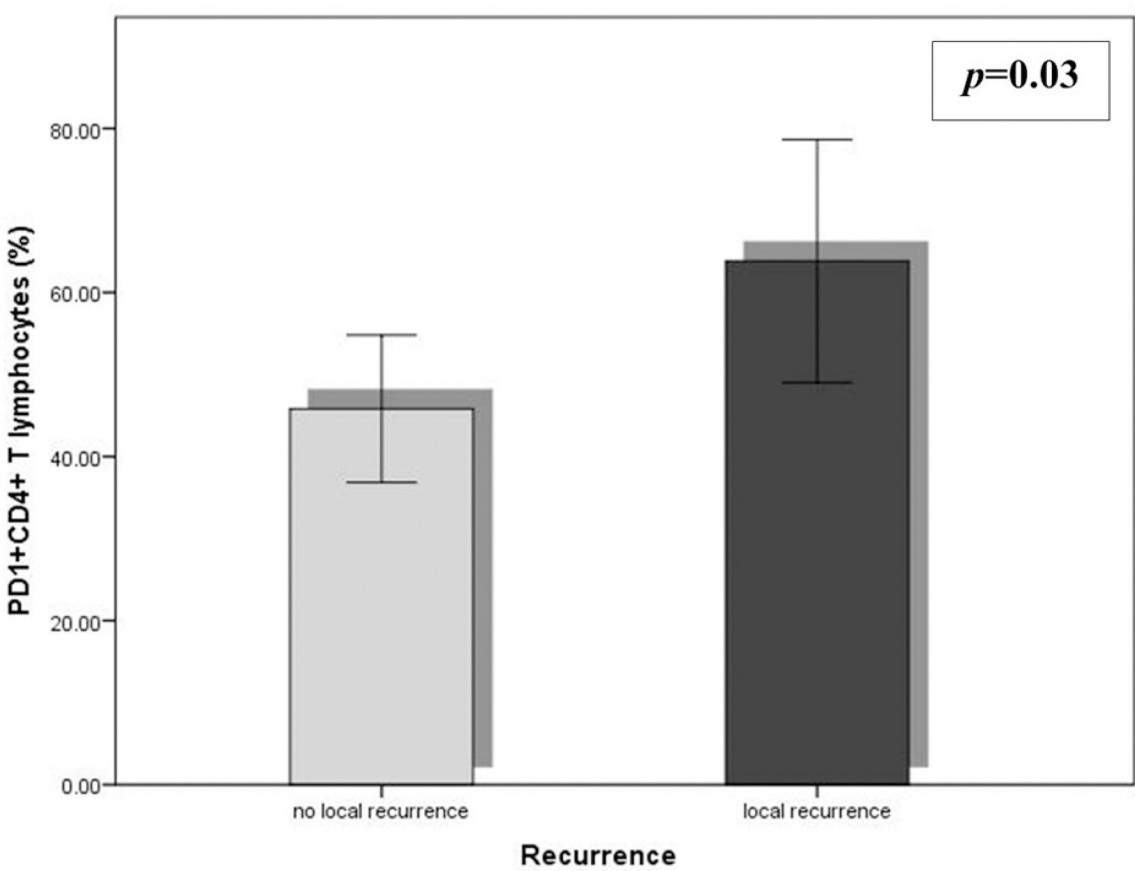

**Fig 4. Differences in the mean percentages of tumor-infiltrating PD1$^+$CD4$^+$ T lymphocytes according to local recurrence ($p$ = 0.03) for the 30 patients with TNBC (Independent t-test).**

## Correlations among tumor-infiltrating and peripheral blood T lymphocyte subsets in patients with triple-negative breast cancer

Correlations between different peripheral and tissue T lymphocyte subsets are demonstrated in Table 5. Peripheral PD1$^+$CD4$^+$T lymphocytes showed positive correlations with both tissue

**Table 3. Differential expression of PD1 and CD39 on peripheral blood T lymphocytes among patients with triple-negative breast cancer and controls.**

| T lymphocyte subsets (%) | Patients (n = 30) | Controls (n = 20) | $p$- value |
|---|---|---|---|
| T lymphocytes | 60.8±2 | 69.1±2 | **0.007** |
| CD4$^+$ | 40.1±1 | 51.3±1 | **<0.0001** |
| CD8$^+$ | 19.9±1 | 16.7±1 | **0.03** |
| PD1$^+$CD4$^+$ | 30.9±3 | 11.6±0.6 | **<0.0001**[a] |
| PD1$^+$CD8$^+$ | 29.6±2 | 13.4±1 | **<0.0001**[a] |
| CD39$^+$CD4$^+$ | 18.3±2 | 8.1±1 | **<0.0001**[a] |
| CD39$^+$CD8$^+$ | 11.2±0.7 | 10.4±0.6 | 0.4[a] |

The percentages of the CD4$^+$ and CD8$^+$ cells were calculated within the T lymphocytes, and the percentages of PD-1$^+$and CD39$^+$ cells were assessed within each of the CD4$^+$ and CD8$^+$ T lymphocytes. Data expressed as mean ±SE, Independent samples t-test,

[a]; Mann-Whitney test for significance, a $p$-value is significant if < 0.05.

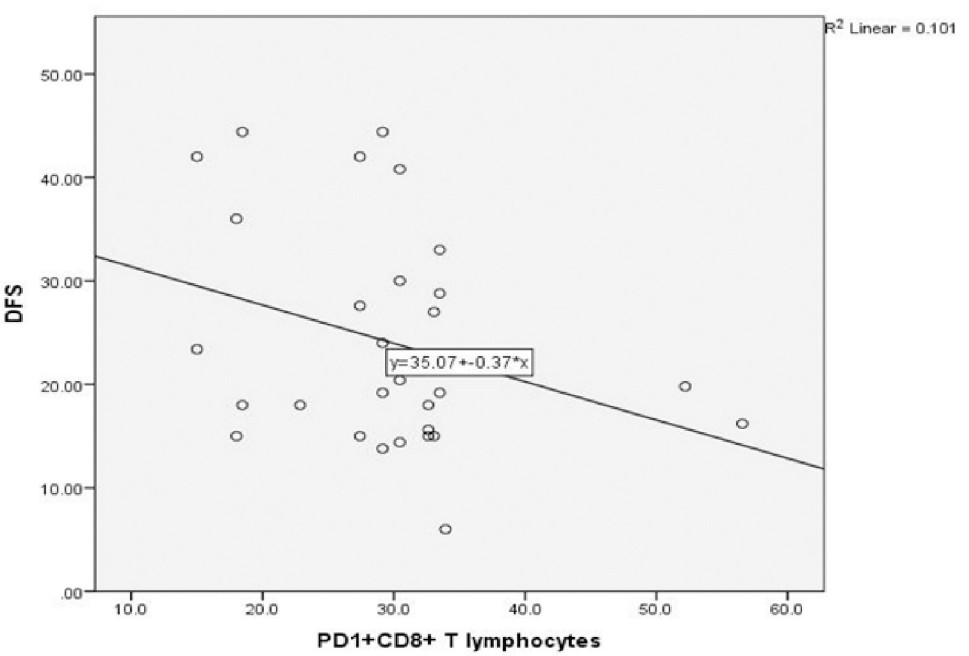

**Fig 5. Correlation of peripheral PD1⁺CD8⁺ T lymphocytes with DFS in 30 TNBC patients ($p$ = 0.04).**

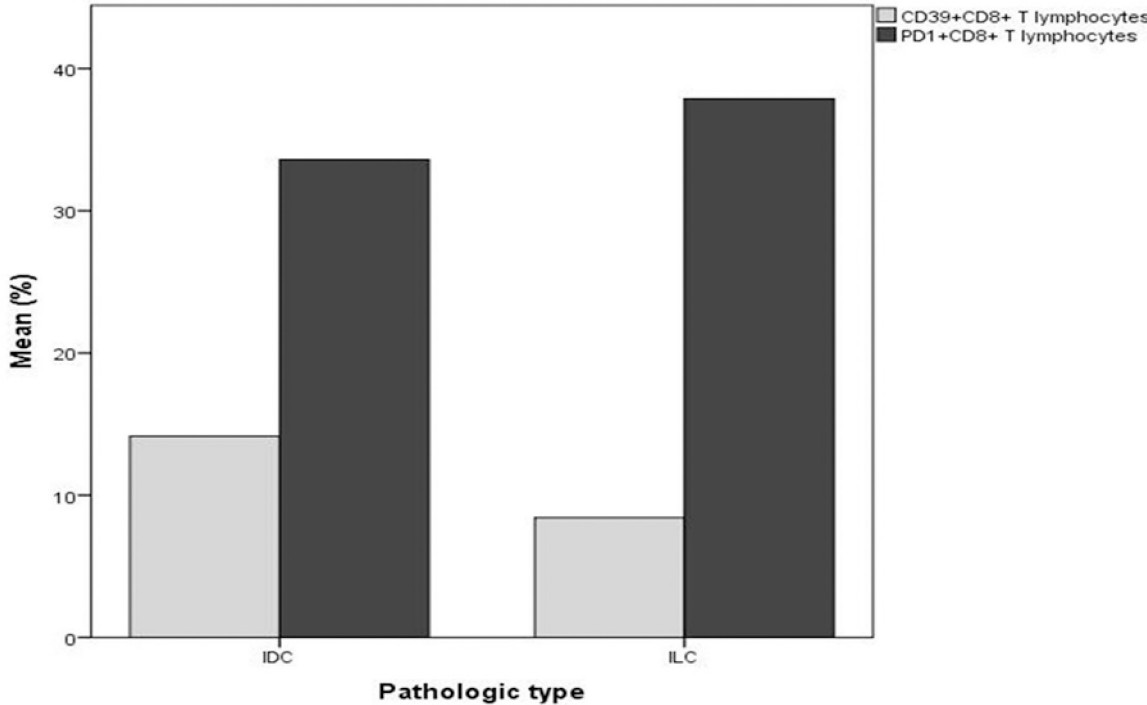

**Fig 6. Significant differences in the mean percentages of PD1⁺CD8⁺ ($p$ = 0.03) and CD39⁺CD8⁺ peripheral T lymphocytes ($p$ = 0.04) according to pathologic type (ILC, and IDC respectively), in 30 patients with TNBC (Independent t-test).**

**Table 4. Differential expression of PD1 and CD39 among tumor-infiltrating and peripheral blood lymphocytes of patients with triple-negative breast cancer.**

| T lymphocyte subsets (%) | Tissue lymphocytes | Peripheral lymphocytes | *p*-value |
|---|---|---|---|
| T lymphocytes | 26.6±0.9 | 60.8±2 | **<0.0001** |
| CD4$^+$ | 58.2±3 | 40.1±1 | **<0.0001** |
| CD8$^+$ | 34.2±3 | 19.9±1 | **<0.0001** |
| PD1$^+$CD4$^+$ | 50.8 ± 4 | 30.9±3 | **<0.0001**[b] |
| PD1$^+$CD8$^+$ | 34.9 ±2 | 29.6 ± 2 | **0.02**[b] |
| CD39$^+$CD4$^+$ | 21±2 | 18.3±2 | **<0.0001**[b] |
| CD39$^+$CD8$^+$ | 13.6±0.9 | 11.2±0.7 | **<0.0001**[b] |

The percentages of the CD4$^+$ and CD8$^+$cells were calculated within the T lymphocytes, and the percentages of PD-1$^+$and CD39$^+$cells were assessed within each of the CD4$^+$ and CD8$^+$ T lymphocytes. Data expressed as mean ±SE, independent sample t-test,
[b]; Wilcoxon signed ranks test, a *p*-value is significant if < 0.05.

CD4$^+$ and CD8$^+$ T lymphocytes expressing PD1 (r = 0.5, *p* = 0.001, and r = 0.4, *p* = 0.01 respectively) and with tissueCD39$^+$CD8$^+$T lymphocytes (r = 0.4, *p* = 0.02). Very strong direct correlations were detected between CD39$^+$CD4$^+$cells in tissue and peripheral blood (r = 0.98, *p*<0.0001) and between CD39$^+$CD8$^+$cells in tissue and peripheral blood (r = 0.9, *p*<0.0001). Peripheral CD4$^+$cells showed negative correlation with tissue CD4$^+$Tcells (r = -0.4, *p* = 0.008), and positive correlation with tissue CD8$^+$T cells (r = 0.4, *p* = 0.008). In the same manner, peripheral CD8$^+$cells showed negative correlation with tissue CD4$^+$Tcells (r = -0.5, *p* = 0.002), and positive correlation with tissue CD8$^+$T cells (r = 0.5, *p* = 0.002).

## Cox regression analysis of flow data on DFS

To evaluate the effects of these lymphocyte subsets simultaneously, that gave significant effect on DFS in univariate analysis, cox regression analysis with enter methods was done to detect the hazard of death from these lymphocytes, from the output below, these lymphocyte covariates were all associated with HR>1; increasing their values were associated with increasing the hazard of death, or local recurrence, especially for CD8$^+$CD39$^+$ T cells, were increasing its

**Table 5. Correlations between tumor tissue and peripheral T lymphocyte subsets in patients with triple-negative breast cancer.**

| Tumor tissue Peripheral blood | | CD4$^+$ | CD8$^+$ | PD1$^+$CD4$^+$ | PD1$^+$CD8$^+$ | CD39$^+$CD4$^+$ | CD39$^+$CD8$^+$ |
|---|---|---|---|---|---|---|---|
| CD4$^+$ | r | **-0.4** | **0.4** | -0.2 | -0.09 | -0.08 | -02 |
| | p | **0.008** | **0.008** | 0.1 | 0.3 | 0.3 | 0.1 |
| CD8$^+$ | r | **-0.5** | **0.5** | 0.08 | -0.06 | -0.2 | -0.1 |
| | p | **0.002** | **0.002** | 0.4 | 0.4 | 0.1 | 0.2 |
| PD1$^+$CD4$^+$ | R | 0.06 | -0.05 | **0.5** | **0.4** | -0.1 | **0.4** |
| | P | 0.4 | 0.4 | **0.001** | **0.01** | 0.3 | **0.02** |
| PD1$^+$CD8$^+$ | R | 0.2 | -0.2 | 0.3 | 0.3 | -0.02 | 0.2 |
| | P | 0.2 | 0.2 | 0.06 | 0.05 | 0.5 | 0.1 |
| CD39$^+$CD4$^+$ | R | 0.08 | -0.08 | -0.1 | -0.1 | **0.98** | -0.1 |
| | P | 0.3 | 0.3 | 0.3 | 0.3 | **<0.0001** | 0.2 |
| CD39$^+$CD8$^+$ | R | 0.1 | -0.09 | 0.2 | -0.1 | -0.1 | **0.9** |
| | P | 0.2 | 0.3 | 0.2 | 0.2 | 0.2 | **<0.0001** |

r; correlation coefficient, a significant *p*-value is <0.05, The percentages of the CD4$^+$ and CD8$^+$ cells were calculated within the T lymphocytes, and the percentages of PD-1$^+$and CD39$^+$ cells were assessed within each of the CD4$^+$, and CD8$^+$ T lymphocytes.

**Table 6. Cox regression analysis of DFS of 30 females with TNBC.**

|  | B | SE | Wald | Sig. | HR | 95% CI for HR | |
|---|---|---|---|---|---|---|---|
|  |  |  |  |  |  | Lower | Upper |
| pPD1+CD8 | .032 | .026 | 1.509 | .219 | 1.033 | .981 | 1.088 |
| CD8CD39 | .050 | .053 | .902 | .342 | 1.052 | .948 | 1.167 |
| PD1+CD8 | .007 | .022 | .096 | .756 | 1.007 | .964 | 1.052 |

SE; standard error, CI; confidence interval, Wald test gives an idea about the importance of the contribution of each variable in the model (corresponding to t-test), the larger value, the greater importance of its specified variable, from the above table peripheral PD1CD8 has the greatest importance on the hazard of death, $HR = \beta o + x1\beta 1 +—-xp\beta p$, p; peripheral.

value by one point increasing the hazard of death or recurrence by 5.2%, this was followed by peripheral PD1+CD8+T cells (HR increased by 3.3%), and at last tissue PD1+CD8+T cells (HR increased by 0.7%), subsequently considered bad prognostic factors as demonstrated in Table 6, and Fig 7.

## Discussion

Although TNBC still represents a minority of breast cancers [24], the significant challenge facing clinical practice is that TNBC is more aggressive, with limited therapeutic options and poor clinical outcomes compared with the non-TNBC [25].

Analysis of CD4+ andCD8+ T lymphocytes and their expression levels of PD and CD39 in TNBC patients have shown several significant findings. Levels of peripheral CD8+ T lymphocytes, including those expressing PD1 and CD39, were significantly elevated in patients compared with controls, and their frequencies in breast cancer tissue were higher than in the

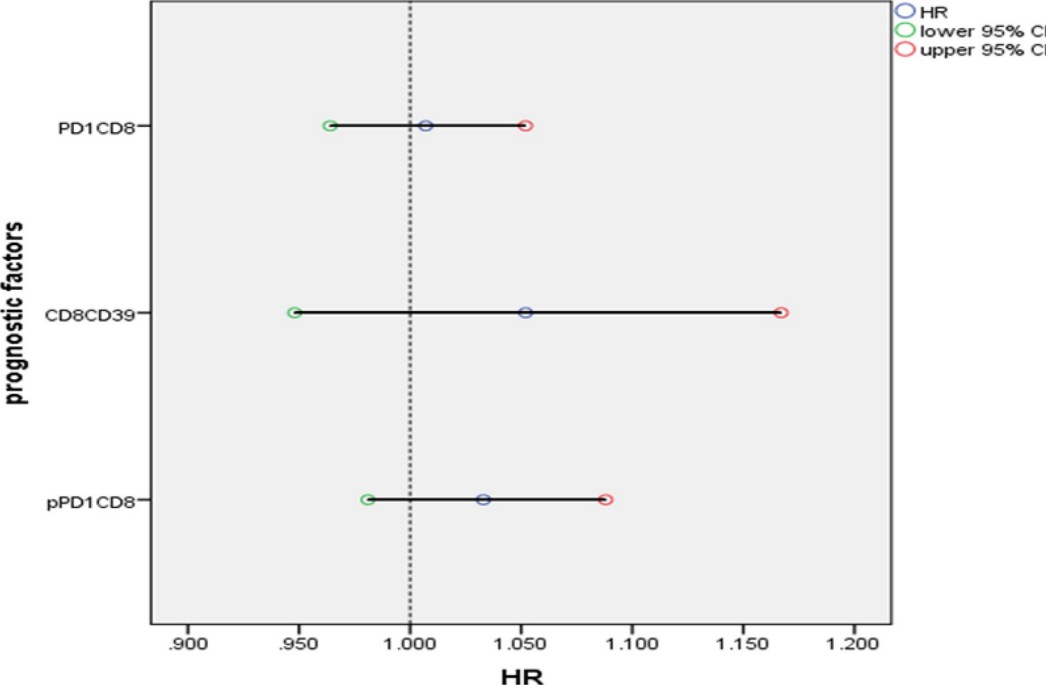

**Fig 7. Forest plot of HR for different lymphocyte subsets.**

periphery. Even though the level of CD8[+] T lymphocytes in breast cancer tissue was lower than that in the healthy tissue, a more significant percentage of cancer tissue CD8[+] T lymphocytes were expressing PD1 and CD39.

Although peripheral CD4+ T lymphocytes were lower in patients than in controls, the percent of PD1 and CD39 expression among these cells was higher in patients. The frequency of CD4[+] T lymphocytes in breast cancer tissue was higher than in the periphery and higher than in healthy tissue. Moreover, more significant percentages of CD4[+] T lymphocytes expressing PD1 and CD39 were detected in breast cancer tissue than in peripheral blood of TNBC and higher than in healthy breast tissue.

Our findings collectively indicate robust trials of the immune system to mount sufficient tumor-specific immune response, possibly by inducing CD4[+] migration and accumulation in breast cancer tissue. Yet, although CD8[+] cell levels increased in the periphery of patients, to limit this antitumor activity, breast cancer probably induced the up-regulation of immune-suppressive molecules as PD1 and CD39 by both CD4[+] and CD8[+] lymphocytes and reduction in the levels of CD8[+] cells in breast cancer tissue. This makes a large proportion of these cells of the terminally exhaustive phenotype to promote tumor growth and hence metastasis.

A substantial increase of TIL represents a robust intra-tumoral inflammatory response with enhanced innate and adaptive immunity and has been linked with a more favorable prognosis in TNBC and promising treatment responses [26–28]. However, the composition of these infiltrating immune cells plays an essential role in disease prognosis [29]. The dominance of infiltrated CD4[+] T lymphocytes and lower levels of CD8[+] T cells in breast cancer tissue have been associated with poor survival [30]. Our results also showed that lower levels of peripheral CD4[+] and CD8[+] T cells were associated with higher levels of CD4[+] and lower levels of CD8[+] T cells in cancer tissue.

Accumulation of T cells in tumor-infiltrating leukocytes with an exhaustive phenotype in breast cancer tissue was previously reported [31]. Of all breast cancers, 30% to 50% were found to up-regulate PD-L1 on their cell surface [32]. TNBCs that are more aggressive than hormone receptor-positive showed higher levels of tumor-infiltrating lymphocytes (TILs) and PD-L1 expression [7]. When PD-L1 binds with PD-1 on the surface of T cells, it dampens the activation of the lymphocytes, making them unresponsive against cancer cells allowing the growth and metastatic spread of the tumor [26, 33].

Increased expression of CD39 was linked to poor prognosis in several tumors [34, 35]. High-level expression of CD39 indicates terminal CD8[+] T cells exhaustion [36]. In a different setting than cancer, activated T cells with high CD39 expression were prone to apoptosis in older individuals. They suggested that CD4[+]CD39[+] effector T cells do not develop into long-lived memory cells [37]. CD39 was also related to tumor cell growth, differentiation, invasion, and migration [38, 39].

Previous researches discussed the interplay between CD39 and PD-1 on tumor-infiltrating immune cells in lung and breast cancers [31, 40]. Both studies revealed considerably increased levels of these immunosuppression markers in tumor tissue than normal tissue. Besides the up-regulated expression of CD39 and PD-1, the expanded CD4+ T cells within the breast tumor tissue showed greater levels of CD25, signifying that although they are activated but exhausted and incapable of mounting any tumor-specific immune response [31].

Furthermore, in our study, positive correlations were detected among peripheral PD1[+]CD4[+]T lymphocytes and each of cancer tissue PD1[+]CD4[+], PD1[+]CD8[+], and CD39[+]CD8[+]T cells and among peripheral and cancer tissue CD39[+]CD4[+] and CD39[+]CD8[+] T cells. Also, Tumor-infiltrating PD1[+]CD4[+] T lymphocytes were higher in patients with local recurrence. This indicates the indirect role played by the latter cells in immune suppression mostly through their relationship with the other T cell subsets.

Altogether, our findings support that adenosine and PD1 signaling pathways may be acting simultaneously for immunosuppression and tumor immune-escape. Supporting this, DFS was inversely related to the tissue PD1+CD8+ and CD39+CD8+ T lymphocytes and peripheral PD1+CD8+ T lymphocytes.

Lower levels of peripheral and breast cancer tissue CD39+CD4+ T lymphocyte were detected in G3 TNBC compared with G2. This may be because most of the patients (83.3%) were having G2 breast cancer.

Notably, the type of immune cells and PD-L1 expression in invasive breast cancer varies relatively depending on the subtype [41]. Infiltration of the tumor microenvironment by lymphocytes and myeloid cells influence prognosis and treatment response [42, 43]. One study [44] observed fewer activated CD8+ T cells in invasive breast cancer than in-ductal carcinoma in situ (DCIS), proposing that immune evasion occurs during the transition from DCIS to invasive cancer. Our results revealed that while the mean percentage of PD1+CD8+ T cells was higher in patients with ILC, CD39+CD8+ T cells were higher in the patients with IDC.

## Conclusion

The CD39 and PD1 inhibitory pathways are synergistically utilized by TNBC cells for evading host immune response causing poor survival. Hence, combined immunotherapy blocking these pathways might be a promising treatment strategy in this type of cancers.

## Acknowledgments

**Ethical approval:** All procedures performed in studies involving human participants were in accordance with the ethical standards of the institutional and/or national research committee and with the 1964 Helsinki declaration and its later amendments or comparable ethical standards and was approved by the local ethics committee of the Faculty of Medicine, Assiut University(IRB number: 17300415).

## Author Contributions

**Conceptualization:** Asmaa M. Zahran, Amal Rayan, Mona H. Abdel-Rahim, Omnia El-Badawy.

**Data curation:** Amal Rayan, Mona H. Abdel-Rahim, Omnia El-Badawy.

**Formal analysis:** Amal Rayan, Omnia El-Badawy.

**Funding acquisition:** Wael M. Y. Mohamed, Dalia O. Mohamed.

**Investigation:** Zeinab Albadry M. Zahran.

**Methodology:** Asmaa M. Zahran, Amal Rayan, Zeinab Albadry M. Zahran, Dalia O. Mohamed, Mona H. Abdel-Rahim, Omnia El-Badawy.

**Resources:** Zeinab Albadry M. Zahran, Wael M. Y. Mohamed, Mona H. Abdel-Rahim.

**Supervision:** Asmaa M. Zahran, Omnia El-Badawy.

**Writing – original draft:** Amal Rayan, Mona H. Abdel-Rahim.

**Writing – review & editing:** Asmaa M. Zahran.

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
