## [Decision Letter · Decision Letter 0]

16 Jul 2021

PONE-D-21-14265

Overexpression of PD-1 and CD39 in tumor-infiltrating lymphocytes compared with peripheral blood lymphocytes in triple-negative breast cancer

PLOS ONE

Dear Dr. Rayan,

Thank you for submitting your manuscript to PLOS ONE. After careful consideration, we feel that it has merit but does not fully meet PLOS ONE’s publication criteria as it currently stands. Therefore, we invite you to submit a revised version of the manuscript that addresses the points raised during the review process.

We look forward to receiving your revised manuscript.

Kind regards,

Sushanta K Banerjee, PhD

Academic Editor

PLOS ONE

Journal Requirements:

Reviewers' comments:

Reviewer's Responses to Questions

**Comments to the Author**

1. Is the manuscript technically sound, and do the data support the conclusions?

Reviewer #1: Yes

2. Has the statistical analysis been performed appropriately and rigorously? 

Reviewer #1: Yes

3. Have the authors made all data underlying the findings in their manuscript fully available?

Reviewer #1: Yes

4. Is the manuscript presented in an intelligible fashion and written in standard English?

Reviewer #1: Yes

5. Review Comments to the Author

Reviewer #1: In this manuscript (PONE-D-21-14265) by Asmaa M. Zahran et al, had investigated the expression of PD-1 and CD39 on CD4+ and CD8+ cells infiltrating tumor tissue compared to their counterparts in peripheral blood. They also explored their association with tumor characteristics, disease progression, and prognosis in females with TNBC. My specific comments are as follows-

1. The authors must mention how they have normalized the flow data for the patient and healthy subset.

2. The single cell suspension protocol from surgically resurrected tumors should be elaborated and properly cited.

3. The authors have shown that the level of peripheral CD4+ T lymphocytes was lower in patients than in controls. An observation by Bushra Sikandar et al contradicts this observation. https://www.ncbi.nlm.nih.gov/pmc/articles/PMC5648386/ . is there any reason that this patient subset behaved differently ?

4. In the discussion section, the authors quote “Although the level of peripheral CD4+ T lymphocytes was lower in patients than in controls, the percent of PD1 and CD39 expression among these cells was higher in patients. The frequency of CD4+ T lymphocytes in breast cancer tissue was higher than in the periphery and higher than that in the healthy tissue.” This statement is self contradictory.

5. Overall the images do not look well organized and could be improved by including multiple graphs in one figure.

6. The manuscript needs formatting and multiple typological errors were detected that could be taken care of.

7. The experimental design is robust and the data supports their hypothesis that the CD39 and PD1 inhibitory pathways are synergistically utilized by TNBC cells for evading host immune response causing poor survival, and hence combined immunotherapy blocking these pathways might be a promising treatment strategy in this type of cancers.

6. PLOS authors have the option to publish the peer review history of their article (what does this mean?). If published, this will include your full peer review and any attached files.

Reviewer #1: No

---

## [Author Response · Author response to Decision Letter 0]

24 Aug 2021

Dear reviewers

Thank you for your timely thorough revisions of our manuscript, we answered your questions and made the corresponding changes in the manuscript, all changes were yellow highlighted.

The authors must mention how they have normalized the flow data for the patient and healthy subset.

Normalization of flow data was done by using natural logarithms of their values, however we repeated their statistical analysis using the corresponding non-parametric tests as declared in the section of statistics. 

The single cell suspension protocol from surgically resurrected tumors should be elaborated and properly cited.

We added reference for the detailed steps of preparation in the methods

The authors have shown that the level of peripheral CD4+ T lymphocytes was lower in patients than in controls. An observation by Bushra Sikandar et al contradicts this observation. https://www.ncbi.nlm.nih.gov/pmc/articles/PMC5648386/ . is there any reason that this patient subset behaved differently ?

Our findings are not contradictory to the above mentioned that of Bushra Sikandar et al. Although level of peripheral CD4+ T lymphocytes in our study was lower in patients than in controls, the frequency of CD4+ T lymphocytes in breast cancer tissue was higher than in the periphery and higher than that in the healthy tissue. Bushra Sikandar et al. studied the percentages of lymphocyte subsets including CD4 lymphocytes only in tissue by immunohistochemistry. They compared the percentages of CD4+ T lymphocytes between breast cancer tissue between patients with TNBC and non-TNBC patients and their results showed significantly increased tumour infiltration of lymphocytes (T and B-lymphocytes) in TNBC compared to the patients with non-TNBC. 

In the discussion section, the authors quote “Although the level of peripheral CD4+ T lymphocytes was lower in patients than in controls, the percent of PD1 and CD39 expression among these cells was higher in patients. The frequency of CD4+ T lymphocytes in breast cancer tissue was higher than in the periphery and higher than that in the healthy tissue.” This statement is self-contradictory.

In our study we compared the level of CD4 and CD8 lymphocytes and their expression of PD1 and CD39 between patients with TNBC and healthy control, both in peripheral blood and breast tissue. level of peripheral CD4+ T lymphocytes in our study was lower in patients than in controls. On the contrary, the frequency of CD4+ T lymphocytes in breast cancer tissue was higher than that in the healthy tissue and was even higher than that in the periphery. We explained this in the discussion “Our findings collectively indicate robust trials of the immune system to mount sufficient tumor-specific immune response, possibly by inducing CD4+ migration and accumulation in breast cancer tissue”.

Overall the images do not look well organized and could be improved by including multiple graphs in one figure.

Three figures were joined in one figure (figures 3A, B, and 4)

The manuscript needs formatting and multiple typological errors were detected that could be taken care of.

Done as much possible

The experimental design is robust and the data supports their hypothesis that the CD39 and PD1 inhibitory pathways are synergistically utilized by TNBC cells for evading host immune response causing poor survival, and hence combined immunotherapy blocking these pathways might be a promising treatment strategy in this type of cancers.

The predominant systemic therapy for most metastatic TNBC is chemotherapy, but responses are often short-lived, and patients have a median overall survival of 12 to 18 months. Therefore, improved therapies are urgently needed.

Immunotherapy has prolonged survival in many solid tumors and represents a promising treatment strategy for TNBC. The most successful immunotherapeutic agents consist of immune checkpoint inhibitors, such as cytotoxic T lymphocyte antigen-4 (CTLA-4) and PD-1 inhibirors, to improve the cytotoxicity and proliferative capacity of tumor-infiltrating lymphocytes. Immune checkpoint inhibitors, including monoclonal antibodies against PD-1 (ie, pembrolizumab, nivolumab), PD-L1 (ie, atezolizumab, durvalumab, avelumab), and CTLA-4 (ie, ipilimumab), have generated durable responses across many tumor types.

---

## [Editor Report · Decision Letter 1]

4 Jan 2022

Overexpression of PD-1 and CD39 in tumor-infiltrating lymphocytes compared with peripheral blood lymphocytes in triple-negative breast cancer

PONE-D-21-14265R1

Dear Dr. Rayan,

We’re pleased to inform you that your manuscript has been judged scientifically suitable for publication and will be formally accepted for publication once it meets all outstanding technical requirements.

Kind regards,

Sushanta K Banerjee, PhD

Academic Editor

PLOS ONE
---

## [Editor Report · Acceptance letter]

10 Jan 2022

PONE-D-21-14265R1 

Overexpression of PD-1 and CD39 in tumor-infiltrating lymphocytes compared with peripheral blood lymphocytes in triple-negative breast cancer 

Dear Dr. Rayan:

I'm pleased to inform you that your manuscript has been deemed suitable for publication in PLOS ONE. Congratulations! Your manuscript is now with our production department. 

Kind regards, 

on behalf of

Professor Sushanta K Banerjee 

Academic Editor

PLOS ONE